

# The role of polycarbonate monomer bisphenol-A in insulin resistance

Milos Pjanic

Department of Medicine, Division of Cardiovascular Medicine, Cardiovascular Institute, Stanford University School of Medicine, Stanford, CA, United States of America

Corresponding author
Milos Pjanic, mpjanic@stanford.edu

## ABSTRACT

Bisphenol A (BPA) is a synthetic unit of polycarbonate polymers and epoxy resins, the types of plastics that could be found in essentially every human population and incorporated into almost every aspect of the modern human society. BPA polymers appear in a wide range of products, from liquid storages (plastic bottles, can and glass linings, water pipes and tanks) and food storages (plastics wraps and containers), to medical and dental devices. BPA polymers could be hydrolyzed spontaneously or in a photo- or temperature-catalyzed process, providing widespread environmental distribution and chronic exposure to the BPA monomer in contemporary human populations. Bisphenol A is also a xenoestrogen, an endocrine-disrupting chemical (EDC) that interferes with the endocrine system mimicking the effects of an estrogen and could potentially keep our endocrine system in a constant perturbation that parallels endocrine disruption arising during pregnancy, such as insulin resistance (IR). Gestational insulin resistance represents a natural biological phenomenon of higher insulin resistance in peripheral tissues of the pregnant females, when nutrients are increasingly being directed to the embryo instead of being stored in peripheral tissues. Gestational diabetes mellitus may appear in healthy non-diabetic females, due to gestational insulin resistance that leads to increased blood sugar levels and hyperinsulinemia (increased insulin production from the pancreatic beta cells). The hypothesis states that unnoticed and constant exposure to this environmental chemical might potentially lead to the formation of chronic low-level endocrine disruptive state that resembles gestational insulin resistance, which might contribute to the development of diabetes. The increasing body of evidence supports the major premises of this hypothesis, as exemplified by the numerous publications examining the association of BPA and insulin resistance, both epidemiological and mechanistic. However, to what extent BPA might contribute to the development of diabetes in the modern societies still remains unknown. In this review, I discuss the chemical properties of BPA and the sources of BPA contamination found in the environment and in human tissues. I provide an overview of mechanisms for the proposed role of bisphenol A in insulin resistance and diabetes, as well as other related diseases, such as cardiovascular diseases. I describe the transmission of BPA effects to the offspring and postulate that gender related differences might originate from differences in liver enzyme levels, such as UDP-glucuronosyltransferase, which is involved in BPA processing and its elimination from the organism. I discuss the molecular mechanisms of BPA action through nuclear and membrane-bound ER receptors, non-monotonic dose response, epigenetic modifications of the DNA and propose that chronic exposure to weak binders, such as BPA, may mimic the effects of strong binders, such as estrogens.

## INTRODUCTION

Bisphenol A (BPA, IUPAC ID: 4,4′-(propane-2,2-diyl)diphenol) is one of the most extensively used synthetic monomers that in a polymerized state constitutes polycarbonate plastics and epoxy resins and makes up the significant part of the plastic environment that surrounds modern human species. Bisphenol A is also a molecule that behaves as an endocrine-disrupting chemical (EDC) that is mimicking the effects of an estrogen (i.e., a xenoestrogen molecule). BPA could interfere with the endocrine system and promote a chronic imbalance that parallels endocrine disruption that arises during pregnancy under the influence of estrogens, such as insulin resistance (IR).

Human populations are being continuously exposed to bisphenol A to the extent that it could be considered a common environmental factor present since 1957, when the first production of BPA started. With over six billion pounds (2.7 million tonnes) of BPA produced in 2003 and incorporated into polycarbonate polymers, and estimated 4.5 million tonnes in 2015, BPA represents one of the most abundant chemicals that come in direct contact with human populations world-wide (*Welshons, Nagel & Vom Saal, 2006*). The volume of BPA production in the world is predicted to increase continuously, and it is presently estimated to surpass 5.4 million tonnes (*Merchant Research & Consulting Ltd, 2017*). BPA is found in plastic bottles, plastic food containers, baby and water bottles, can and glass linings, various medical and dental devices, sealants for dental fillings, compact disks and electronics, eyeglass lenses, and even in the lining of water pipes and tanks (*Talsness et al., 2009*; *Huang et al., 2012*). BPA is the main monomer of epoxy resins which are being used as coating agents on the interior of many water storage tanks. Hence, BPA leaching from such widely used polymers may influence human health inadvertently through consuming water or food. In addition, canned food might represent one of the significant global sources of BPA, as it has been shown that the canned food is significantly more contaminated with BPA compared to the non-canned one and that the BPA levels from canned food did not differ between continents (*Repossi et al., 2016*). In addition to dietary ingestion, a significant source of BPA exposure in modern human populations represents thermal paper used for supermarket and ATM receipts, that could efficiently transfer BPA to skin when holding the source for as long as 5 s, and it remains largely extractable after 2 h, indicating perfusion into the skin to such depths that it could no longer be removed or washed off easily (*Biedermann, Tschudin & Grob, 2010*; *Ehrlich et al., 2014*). Furthermore, one of the sources that could provide constant exposure to BPA is orthodontic material, as it has been shown that the eluents from orthodontic adhesives immersed in saline for one month at 37 °C exhibit estrogenic action through the induction of MCF-7 cell proliferation, an estrogen-responsive cell line (*Gioka et al., 2009*).

In addition, BPA is found as an additive to various polyvinyl chloride (PVC) products, including food packaging stretch films (*López-Cervantes & Paseiro-Losada, 2003*). Food and

drink products are estimated to be the major source of BPA contamination, with detected concentrations reaching microgram (µg) levels. BPA and several bisphenol A analogues, including bisphenol-F (BPF) and bisphenol-S (BPS), have been found in various categories of food and drink products including fruits, vegetables, dairy products, cereals, beverages, fats and oils, fish and seafood, meat and meat products and "other products" category using high-performance liquid chromatography-tandem mass spectrometry (HPLC-MS/MS) (*Liao & Kannan, 2013*). BPA was detected in 75% of the tested samples, and the detected concentration was ranging from the very limit of quantification (LOQ) to 1.13 µg/g weight (mean value, 4.38 ng/g). BPA and its analogues have been shown to appear in the indoor dust, and dust ingestion may be another significant source of contamination with BPA (*Liao et al., 2012*). The potential to use BPA analogues, BPS and BPF, as substitutes in the production of plastics has been debated, as BPA analogues have shown similar effects to BPA on the cellular level (*Verbanck et al., 2017*).

The global industrial trend of polycarbonate consumption is increasing from 1.6 million tonnes in 2000 up to 3.4 million tonnes in 2010, and 4.5 million tonnes estimated in 2015 (*Sevak Publications, 2008*). Significant demand of BPA originates from East Asia (predominantly China), that accounted for 59% of the polycarbonate consumption in 2010 and 68% estimated in 2015. On the other side, Europe contributed with 32% of polycarbonate consumption in 2000, 30% in 2010, and estimated 25% in 2015. Similar decreasing trend was observed for the United States (28% in 2000, 22% in 2010, and 18% in 2015). Particularly, the production of BPA in China has grown extensively and this continuous trend will lead to the increasing BPA contamination in the global environment (*Huang et al., 2012*). The Environmental Protection Agency (EPA) in the US has established the tolerable daily intake (TDI) for BPA at 50 µg per kg (body weight) per day in 1988 (*Rubin, 2011*). In January 2015, European Food Safety Authority has reduced TDI for BPA from 50 to 4 µg per kg (body weight) per day. In addition, EPA has established the oral reference dose (RfD) for BPA at 100 µg/L as a standard for the total allowable concentration (TAC) in drinking water. Whether the extent of such exposure is significant has been debated (*Nohynek et al., 2013*), and various evidence have been presented supporting that the tight regulation of BPA is necessary. This is especially important considering the new knowledge about BPA action, including the numerous instances of experimental non-monotonic dose response to the BPA treatment, indicating narrow effective range of concentrations and the absence of effects on higher doses.

In this review, I present a global overview of the chemical properties of BPA and its polymers, their hydrolytic reaction and leaching BPA concentrations present in the environment and the corresponding concentrations in human tissues. Next, I define the physiological gestational insulin resistance (GIR), its pathophysiological progression to gestational diabetes and the role of estrogens in promoting these disorders. I elaborate on the proposed mechanism of BPA endocrine disruption, including non-monotonic dose response, and its transgenerational effects on male and female offspring, including epigenetic modifications transmitted to the next generation. Finally, I review the literature on biological effects of BPA in mice and humans including insulin resistance and diabetes,

as well as in cardiovascular diseases, such as coronary artery disease and other physiological disorders that BPA might contribute to.

## SURVEY METHODOLOGY

In order to survey the effects of BPA on insulin resistance I searched for studies analyzing BPA and insulin resistance on PubMed or PubMed Central (PMC) from their inception through May 2, 2017 using the following search algorithm: (*bpa[Title/Abstract] OR bisphenol-A[Title/Abstract]) AND ("insulin resistance"[MeSH Terms] OR ("insulin"[Title/Abstract] AND "resistance"[Title/Abstract]) OR "insulin resistance"[Title/Abstract])*. This search yielded the list of 19 publications on PubMed Central and 86 publications on PubMed. The survey of papers describing the influence of BPA on cardiovascular diseases was performed by applying the search algorithm: (*bpa[Title/Abstract] OR bisphenol-A[Title/Abstract]) AND "cardiovascular disease*"[Title/Abstract],* which yielded 33 publications on PubMed and no publications on PMC. The search with algorithm: (*bpa[Title/Abstract] OR bisphenol-A[Title/Abstract]) AND "cardiovascular"[Title/Abstract]* yielded 97 publications on PubMed and five publications on PMC. The survey of papers describing chemical properties of BPA molecule was performed with the algorithm: (*bpa[Title/Abstract] OR bisphenol-A[Title/Abstract]) AND "chemical properties"[Title/Abstract]*, which yielded 27 publications on PubMed and no items on PMC. The survey of BPA concentration levels in both environment and human tissues was performed with the algorithm: *(bpa[Title/Abstract] OR bisphenol-A[Title/Abstract]) AND "concentration*"[Title]*, yielding 89 publications on PubMed and 12 on PMC. Papers with subject relevant to the search term and not present in the initial search were obtained through Similar Article PubMed function. Initial papers that describe the structural and functional properties of BPA as an estrogen-like molecule from 1936 and 1938 were not present in the PubMed or PMC databases and were found on Wikipedia and obtained from the JStore database. In addition, the search was performed using the '*' symbol that denotes the truncated search terms to increase the number of publications obtained. After reviewing, articles were excluded from the study in case they were published in languages other than English or if they described subjects that were not related to the main search topic. Papers that did not contributed to the scientific understanding of the search topic were excluded, as well as papers that were addressing the same or similar subjects in order to eliminate the redundant studies. After filtering for each search term the number of papers retained were: BPA chemical properties—6, BPA and insulin resistance relationship—11, BPA and cardiovascular and other diseases—4, and BPA and environmental and human tissue concentrations—18. Additional papers were introduced after the revision process.

### BPA chemical properties, polymerization and hydrolysis

Bisphenol A (BPA), is an organic synthetic molecule composed of the two hydroxy-phenyl groups connected through a carbon atom and, therefore, belonging to the group of diphenylmethane derivatives, with the formula $(CH_3)_2C(C_6H_4OH)_2$. BPA holds certain steric resemblance to the estrogen molecule 17β-estradiol (E2), especially in the span of

outer hydroxyl groups, and although it does not have the structure of a complete steroid ring, its behavior as a synthetic estrogen is based on the similar chemical properties, steric mimicking of an estrogen ring and on a weak interaction with the estrogen receptors. BPA has an average mass of 228.3 Da, while estrogen E2 has an average mass 272.4 Da (*Pence & Williams, 2010*). BPA possesses experimental melting point of 153−158 °C (Alfa Aesar), while E2 has a similar, but higher value 175−178 °C (Alfa Aesar, Haverhill, MA, USA). Acidic dissociation constant, $K_a$, for BPA is 10.29 and for E2 is 10.27 (*Bento et al., 2014*). BPA cross-linking properties have propelled its utilization in the manufacturing of polycarbonate plastics and epoxy resins. The polymer structural properties and efficiency of polymerization and degradation when exposed to higher than normal temperatures are essential for the degree of contamination in the environment. The glass-liquid transition temperature (Tg, in amorphous materials represents transition from a compact glassy state into a viscous state) of BPA polycarbonate polymers is 147 °C, while heat deflection temperature (defining polymer deformation under a specified load) is 128 °C under 1.8 MPa. Direct photochemical effect on BPA involves irreversible photo-scission leading to bisphenol-like products (*Rivaton, Sallet & Lemaire, 1983*; *Collin et al., 2012*) and only secondary photo-reactions are influenced by oxygen and may involve Photo-Fries rearrangement of the benzyl groups. On the other side, photo aging of the BPA polycarbonate has been shown also to occur through ring oxidation, e.g., resin was photo-oxidized under both sunlight ($A > 300$ nm) and Hg arc light ($A > 280$ nm) which indicated the loss of aromatic groups (*Clark & Munro, 1984*). Capillary gas chromatogram showed BPA to be highly prevalent in the photo-aged polycarbonate product mixture (*Factor, Ligon & May, 1987*). BPA, therefore, has the potential of leaching out from the food and liquid storage units as well as from the medical and dental materials, especially if exposed to higher temperatures or through a photo-oxidation mechanism.

## BPA exposure levels in human tissues

In the late 90s Japan's can industry has changed its formula for plastic can linings, which has been associated to over 50% decrease in human BPA levels and to the loss of correlation between usage of canned drinks and urine BPA levels in Japan (*Matsumoto et al., 2003*). A wide range of BPA levels has been detected in the adult and fetal serum in humans. Only two studies did not detect any BPA in humans, while in all other studies serum BPA was detected with the concentrations ranging from 0.32 to 4.4 ng ml$^{-1}$ (*Vandenberg et al., 2007*). Several studies testing various human tissues detected even higher BPA concentrations than those present in the serum, with the highest appearing in placenta 11.2 ng ml$^{-1}$ (*Schönfelder et al., 2002*), umbilical cord 4.4 ng ml$^{-1}$ (*Todaka & Mori, 2002*), and amniotic fluid 8.3 ng ml$^{-1}$ (*Ikezuki et al., 2002*), as well as in colostrum (late pregnancy milk) 3.4 ng ml$^{-1}$ (*Kuruto-Niwa et al., 2007*) and breast milk 7.3 ng ml$^{-1}$ (*Ye et al., 2006*). These independent findings are the indication of potential transmission of the effects of elevated BPA exposure from mothers to the progeny, either through placental transfer or breast feeding. The highest concentrations of BPA were found in human saliva immediately after the dental sealant application, 42.8 ng ml$^{-1}$, with the levels dropping to 7.9 ng ml$^{-1}$ 1h after the application (*Joskow et al., 2006*; *Vandenberg et al., 2007*). A recent study found

significant differences in BPA concentrations in saliva between a group of patients with tooth surfaces filled with a polymer-based dental material and a control group without any polymer-based materials ($p = 0.044$, Mann–Whitney $U$ test) (*Berge et al., 2017*). These findings imply a potential long term exposure of BPA after dental surgeries.

In the human urine, BPA was found with detection rates from 52–100% (*Vandenberg et al., 2007*). In the studies from 2005 to 2007, BPA was found in human urine with detection rates of 96% (*Calafat et al., 2005*), 89% (*Liu, Wolff & Moline, 2005*), 97% (*Ye et al., 2005*), 97.5% (*Yang et al., 2006*) and 94% (*Wolff et al., 2007*). These results indicate that BPA in human urine has been almost completely detectable in all tested individuals and confirms a broad human exposure to BPA. Another source of newborn and infant exposure to BPA might be the persistent leaching from the baby bottles. While a study from 1997 failed to detect any traces of BPA in baby bottles (*Mountfort et al., 1997*), a study from 2001 found 2.1 ng ml$^{-1}$ in distilled water that came in contact with the baby bottles for 30 s at 100 °C (*D'Antuono et al., 2001*). Similarly, a study from 2003 found BPA leaching from the baby bottles in concentration of 0.23 ng ml$^{-1}$ in the distilled water after 1 h at 100 °C, as well as increased BPA levels, ranging from 6.7 to 8.4 ng ml$^{-1}$, after repeated cycles of bottle washing and brushing (*Brede et al., 2003*).

A study from 2004, found BPA leaching levels from a polycarbonate tubing to be as high as 3 ng ml$^{-1}$ per day released into the passing water (*Sajiki & Yonekubo, 2004*). More recent study found that BPA was detected in 46.9% of cardboard samples for the take-out food that could potentially be leaching the chemical to the packaged food (*Lopez-Espinosa et al., 2007*). These results indicate that water and food may be the dominant sources of BPA contamination depending on the composition of material used for their packaging and transport.

## Gestational insulin resistance and diabetes mellitus

Gestational insulin resistance (IR) is a naturally occurring physiological phenomenon that arises during pregnancy in order to direct the blood circulating nutrients of pregnant females to the growing fetus (*Mack & Tomich, 2017*). Gestational IR appears as the result of diabetogenic hormonal action originating from placenta, including hormones such as growth hormone, corticotropin-releasing hormone, placental lactogen and progesterone. Subsequently, as a response to increasing insulin resistance in the peripheral tissues, beta-cells of pancreas undergo hypertrophy and start secreting larger amounts of insulin. In certain cases, the inefficacy of larger levels of insulin to compensate for insulin resistance leads to gestational diabetes mellitus (GDM) that manifests itself as increased blood glucose levels in otherwise healthy pregnant females that had not been suffering from diabetes previously (*Chiefari et al., 2017*).

Long-term consequences of gestational diabetes mellitus exist for both mothers and offspring. Even though in most cases the maternal blood glucose levels after pregnancy are brought down to normal levels, women that experiences GDM have higher risk for developing diabetes later in life (*Damm et al., 2016*). In fact, GDM is one of the strongest predictors of diabetes mellitus. It has been estimated that approximately one-third of women that suffer from diabetes mellitus had experienced GDM previously

(*Cheung & Byth, 2003*). Epidemiological studies indicate that the risk of developing type 2 diabetes after suffering from GDM can be largely attenuated by lifestyle intervention and physical activity by 50% (*Buchanan et al., 2002*; *Ratner et al., 2008*). The long-term effects on the offspring of mothers with GDM have been less extensively studied; however, in animal studies it has been shown that offspring of mothers with GDM have increased risk of GDM, diabetes, obesity, cardiovascular disease and dysplasia of the ventromedial hypothalamic nucleus, which is involved in the regulation of metabolism and inhibits insulin secretion (*Harder et al., 2001*; *Aerts & Van Assche, 2006*). In fact, induced lesions in hypothalamic ventromedial nucleus in rats resulted in significant hyperinsulinemia and increased blood glucose levels (*Satoh et al., 1997*). Therefore, malformation of the ventromedial nucleus may be in part responsible for the persisting alterations of glucose homeostasis found in the offspring of mothers with GDM.

GDM is one of the most dominant pregnancy complications as it affects from 2% to 10% of all pregnancies (*Wedekind & Belkacemi, 2016*). During pregnancy, peripheral insulin resistance first appears as a physiological response to changes in steroid balance in the organism (*Vejrazkova et al., 2014*). In fact, the similar effect could be observed with the application of hormonal contraceptives, primarily those containing estrogens, that have been associated with changes in carbohydrate metabolism and increased insulin resistance (*Lopez, Grimes & Schulz, 2012*). For example, one study showed 43–61% increase in plasma glucose levels on the oral glucose-tolerance test (OGTT) in women taking oral contraceptives (*Godsland et al., 1990*), while other studies, in addition to increased OGTT plasma glucose, have found elevated fasting and post-glucose insulin levels and recommended that estrogen content of oral contraceptives should be reduced to minimize the diabetogenic effects (*Wynn et al., 1979*).

## BPA-induced endocrine disruption and insulin resistance

In a 2014 study, it has been demonstrated that the offspring from BPA-exposed mice showed adverse symptoms of diabetes (*García-Arevalo et al., 2014*). In male offspring, the BPA treated group, similar to the groups fed with high fat diet (HFD) and with high fat diet plus BPA, showed fasting hyperglycemia, glucose intolerance and higher levels of insulin and free fatty acids. In 17-week old male offspring, under the fasting conditions, the BPA group of mice was hyperglycemic compared to the control, however this effect of BPA was masked under the high fat diet (HFD vs HFD-BPA). In addition, at 17 weeks, the BPA, HFD and HFD-BPA groups showed higher insulin levels than the control group. Intraperitoneal glucose tolerance test (ipGTT) showed higher glucose intolerance in the HFD and HFD-BPA groups compared to the control, while the BPA group showed a similar tendency without reaching the statistical significance. In 28-week old male offspring, the BPA group had the highest fasting plasma glucose levels and the highest insulin levels, even compared to HFD and HFD-BPA groups. At 28 weeks, ipGTT showed that all three tested groups (BPA, HFD and HFD-BPA) acquired higher glucose intolerance compared to the control. In addition, at 28 weeks, insulin sensitivity, measured by intraperitoneal insulin tolerance test (ipITT), showed high tendency of impairment in BPA, HFD and HFD-BPA groups compared to the control group. Remarkably, the BPA group gained more weight

starting from the 18th week compared to the control, and kept increasing the weight until it reached the levels of the HFD and HFD-BPA groups.

In addition, BPA exposure in mice shows an effect on mothers' glucose metabolism during pregnancy, as well as later in life (*Alonso-Magdalena et al., 2010*). During pregnancy, the treatment with BPA at doses of 10 or 100 µg/kg/day on days of gestation 9-16 (BPA10 and BPA100 mice, respectively) induced a tendency to glucose intolerance measured with ipGTT. In BPA10 mice increased fasting plasma insulin levels were observed (1.38 times higher). In ipITT, both control and BPA10 treated groups, showed only a slight decrease in serum glucose levels, consistent with the physiological insulin resistance in late pregnancy. While BPA10 mice did not alter naturally occurring insulin resistance during pregnancy, in BPA100 mice modestly induced insulin sensitivity was observed. This indicates that BPA10 mice were unable to compensate with their elevated insulin levels to counteract physiological insulin resistance. In addition, in liver BPA treatment reversed insulin-stimulated Akt phosphorylation that occurs in control F0 mice, indicating BPA effects on signaling pathways in liver that are consistent with the strong insulin resistance in these mice. Consequences for mothers later in life included increased body weight four months after delivery, decreased insulin sensitivity measured using ipITT and 2.2 times higher plasma insulin levels after fasting compared to controls. In addition, increased plasma leptin, triglyceride, and glycerol levels were observed in mothers four months after delivery relative to controls. Another study has reported 3% increase in the body weight during the postpartum period and at five and six months postpartum found substantial impairment of glucose tolerance and decreased insulin sensitivity. These alterations in glucose metabolism appeared in pregnant, but not in non-pregnant female mice (*Alonso-Magdalena et al., 2015*). Taken together these data indicate that exposure to BPA during pregnancy produces dysregulated nutrient metabolism later in life. Model animals, therefore, present a valuable source of information on the effect of BPA on insulin resistance, type 2 diabetes and obesity and unveil the connection of environmental estrogens to these phenotypes.

## Molecular mechanisms of BPA in promoting endocrine disruption, gestational insulin resistance and diabetes mellitus

BPA has been first reported to act as a synthetic estrogen in 1936 (*Dodds & Lawson, 1936*), well before scientists discovered that it could be polymerized into polycarbonate plastics in the 1950s. As a xenoestrogen and an endocrine-disrupting chemical, BPA has a potential to intervene with any aspect of the hormonal function, to change the hormonal equilibrium and subsequently affect many physiological processes in different tissues. The mechanism of BPA action as a xenoestrogen is thought to be through binding and competing for estrogen receptors, ER-alpha (ERα) and ER-beta (ERβ) (Fig. 1) (*Dodds & Lawson, 1938*). However, the interaction of BPA with ER receptors is relatively weak, ranging 2 to 3 orders of magnitude lower compared to estrogens (relative recruitment ability—RRA; E2 and BPA recruitment to ERα, 100 and 0.073, respectively; E2 and BPA recruitment to ERβ, 100 and 0.75, respectively) (*Routledge et al., 2000*). Therefore, whether chronic and low-dose BPA exposures function through the ER pathways is still debatable

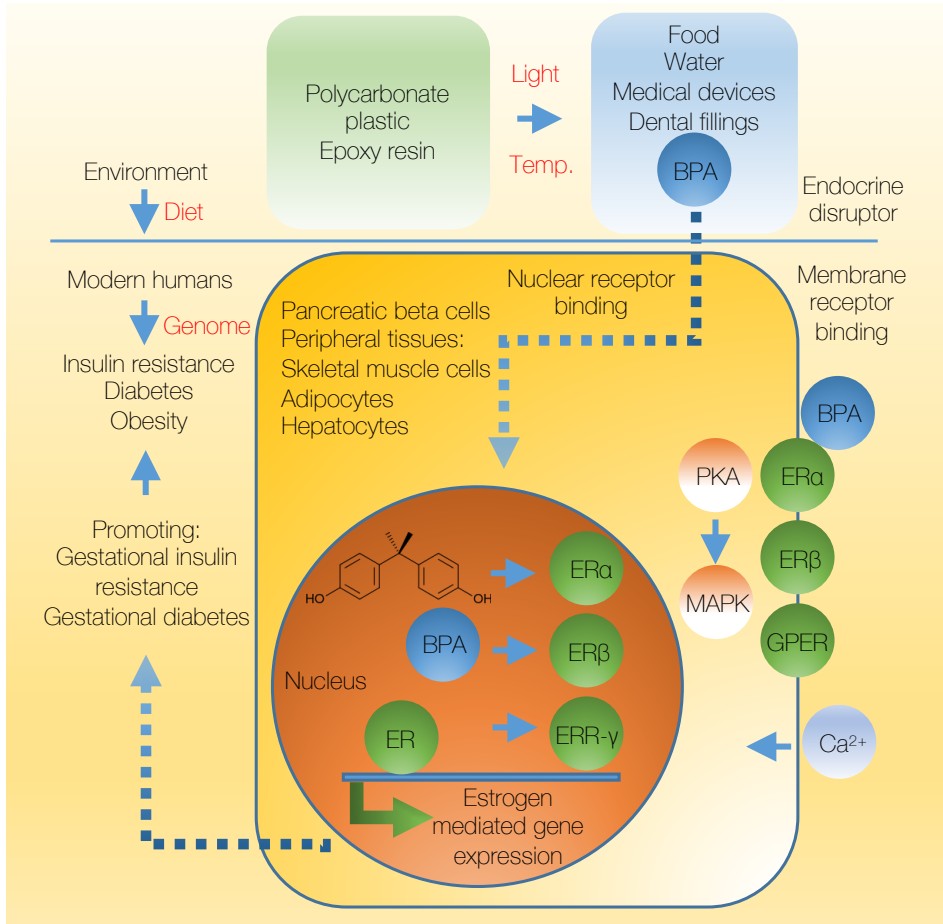

**Figure 1 The model of BPA effect on insulin resistance and diabetes.** Global model of the contribution of endocrine-disrupting chemical BPA to the development of insulin resistance and diabetes in humans. Light and temperature might induce the hydrolysis of polycarbonate plastics and subsequent leaching of BPA into the water and food sources. Once in human tissues BPA exerts its effects through ER receptors alpha, beta and gamma and estrogen mediated gene expression. In addition to the nuclear receptors, BPA can exert its effects through non-genomic, membrane associated receptors. Model indicates that BPA in-duced endocrine disruption may partially contribute to the development of insulin resistance, with major contributors being modern human diet and genomic composition. ER, estrogen receptor. ERR, estrogen-related receptor; GPER, G-protein-coupled estrogen receptor.

(*Safe et al., 2002*). Nevertheless, BPA, as well as other xenoestrogens, are indeed able to efficiently and fully displace radioactively labeled E2 from both ERα and ERβ receptors in a dose dependent manner using a ligand binding assay (*Routledge et al., 2000*). In addition, recent findings indicate that BPA may act through a receptor named estrogen-related receptor gamma (ERR-γ) (*Takayanagi et al., 2006*; *Okada et al., 2008*). ERR-γ is a member of estrogen-related receptor class of genes, a subfamily of orphan nuclear receptors, closely related to the ERs. BPA was found to bind ERR-γ in both a direct receptor binding assays (FRET), as potently as a tracer for ERR-γ, and in a cell-based reporter assay where it rescues high constitutive ERR-γ activity in HeLa cells treated with an ERR-γ inverse agonist 4-hydroxytamoxifen (4-OHT) (*Okada et al., 2008*). Whether BPA exerts its effect

on insulin resistance primarily through one of these mechanisms by mimicking estrogen action remains still to be shown, especially since the risk assessment of xenoestrogens based primarily on the reporter gene and binding assays may be insufficient.

In addition, while BPA is capable of displacing E2 when binding to ERα or ERβ in a dose dependent manner, it has been shown that BPA interacts with a different set of transcriptional co-regulators than E2 (*Routledge et al., 2000*). As ERs regulate target gene transcription by the association with their co-regulators, interaction with a specific set of transcriptional co-regulators may mean BPA binding induces a unique ER conformation and subsequently recruits a unique set of co-regulators conferring specific gene regulation pattern to the target genes. In addition, BPA demonstrated clear preference for certain co-regulators when bound to ERβ compared to ERα. For example, BPA-ERβ complex had 500-fold higher recruitment affinity for coactivator TIF2 than BPA-ERα complex (RRA for BPA-ERα and BPA-ERβ < 0.0001 and 0.05, respectively), which was directionally equivalent to BPA showing almost 10-fold greater binding affinity for ERβ than ERα (RRA for ERα and ERβ 0.073 and 0.75, respectively) (*Routledge et al., 2000*). BPA can therefore confer specific conformational changes to ERβ and enhance ERβ recruitment activity by the order of magnitude compared to its binding effect on ERα. As the main function of E2 binding to ERs is the induction of ER conformational change, in order for the subsequent recruitment of coactivators and assembly of the basal transcription machinery factors, the finding that BPA can efficiently displace E2 from the ER and induce a unique conformational structure of ER (particularly potentiating specific co-factor recruitment to ERβ) and the fact that similar has been shown for other xenoestrogens (*Routledge et al., 2000*), suggests an effective mechanism of BPA action through ER binding.

Other than the proposed ER-activation mechanism that involves binding to nuclear receptors, it has been suggested that BPA may exert its effects through other mechanisms, such as through rapid non-genomic (i.e., non-nuclear) pathways (Fig. 1) (*Welshons, Nagel & Vom Saal , 2006*; *Alonso-Magdalena et al., 2012*). Such systems are cell signaling systems with receptors associated to the membrane and the response time of such systems are generally shorter than those mediated by nuclear receptors (*Judy & Welshons, 2010*). In addition, responsiveness can usually be achieved by smaller initial concentrations of the hormone, indicating high level of signal amplification. In 2000, *Nadal et al. (2000)* demonstrated for the first time that in pancreatic beta cells xenoestrogens, such as BPA, as well as E2, both occupy a common membrane binding site that is distinct from the cytoplasmatic/nuclear ER receptors. Subsequently, it has been shown that BPA at picomolar concentrations can cause rapid influx of calcium ion (within 30 s of administration) followed with a prolactin release in pituitary tumor cells (characterized by the high levels of ERα receptor in the plasma membrane) (*Wozniak, Bulayeva & Watson, 2005*). Evidence suggested that BPA alters the conformation of voltage-dependent calcium channels through membrane depolarization, and the prolactin release was fully preventable by removing calcium from the extracellular solution. Similar was detected in mouse pancreatic cells where BPA treatment promoted calcium influx at the nanomolar concentrations (as low as 1nm), with a depolarization-induced influx subsequently causing CREB phosphorylation (*Quesada et al., 2002*). Cellular responsiveness to lower than

nanomolar BPA concentrations (0.1 nm) was detected also in MCF-7 human breast cancer cells (*Walsh, Dockery & Doolan, 2005*). Similarly, it has been found that the BPA exposure in mouse Leydig cells induced activation of protein kinase A and phosphorylation of MAPK only 5 min after administration and subsequent rapid induction of Nur77 gene expression that regulates testosterone synthesis (Fig. 1), which is an effect that could not be mediated by the prolonged response of nuclear receptors (*Song, Lee & Choi, 2002*). Rapid membrane BPA response was confirmed by protein kinase inhibitor H-89 that strongly inhibited BPA-mediated Nur77 gene activation.

The present concept of BPA estrogenicity (mainly based on nuclear receptor activation) demands substantial refinement to incorporate such rapid responses of estrogen receptors found in the membrane (*Nadal et al., in press*). It has been shown that ERα and ERβ could be expressed in both nuclear and membrane cell fractions and could be functional outside the nucleus (*Razandi et al., 1999*). Transfection of ERα cDNA in CHO cells resulted in the presence of ERα in nuclear and membrane associated cellular fractions and in specific binding of labeled E2 to the CHO cells. The membrane bound ER fractions had similar dissociation constant as nuclear receptors (0.287 and 0.283 nM, respectively), however they represented minor fractions (3% and 2%, ERα and ERβ, respectively) of total receptors in the cell. E2 binding to the membrane ERs resulted in activation of G proteins in the membrane followed by increased adenylate cyclase activity. In 2005, the specific class of steroid membrane receptors named orphan receptor GPR30 (now denoted as G-protein-coupled estrogen receptor, GPER) was discovered in the human breast cancer cells, which could propagate the non-classical estrogen effects localized at the cell surface (*Thomas et al., 2005*). Specific E2 binding was detected in SKBR3 breast cancer cells that express GPER, but are missing estrogen receptors, an effect that was reversible with siRNA-mediated decrease in GPER expression. GPER pre-genomic signaling activity was confirmed to be conducted through the plasma membrane G-protein coupled receptor activities, specifically, the activation of adenylyl cyclase (*Filardo & Thomas, 2012*). GPER knock-out mice are viable, however they show impaired glucose tolerance, potentiating the role of ER non-genomic signaling in glucose homeostasis (*Mårtensson et al., 2009*). In addition, it is speculated that estrogens could bind a large variety of receptors at the plasma membrane surface (*Kow & Pfaff, 2016*) and that their effects almost always reproduce the effects of the estrogenic agents that permeate the cell.

It has been demonstrated that BPA treatment of isolated pancreatic islets, similar to E2, shows the non-monotonic dose response (i.e., an inverted U-shape curve) when insulin content is measured after 48-hour treatment (*Alonso-Magdalena et al., 2008*). Similar response was observed *in vivo* when mice were treated with 100 µg/kg or 1 mg/kg of BPA during 4 days, as low doses of BPA increased insulin content, while the higher doses of BPA exhibited no effect. This non-monotonic dose response was ERα dependent as ERα agonists showed similar pattern of activation. It has been estimated that BPA non-monotonic dose response occurs in 20% of experiments conducted using BPA and in 30% of the literature related to BPA the non-monotonic dose-response has been detected (*Vandenberg, 2014*). The non-monotonic dose response to BPA, while contradicting traditional concepts of toxicology, at the molecular level indeed does not involve BPA acting
as a standard toxic chemical, but rather as a hormonal molecule that initiates signaling cascades through the interaction with a hormone receptor. The molecular mechanisms behind such non-linear relationship between the dose and response of EDC may involve receptor saturation, negative feedback loop mechanisms, and/or non-linear process of receptor homodimerization. It has been shown that computational models can accurately predict the appearance of non-monotonic dose response in gene expression within the standard genomic niche of E2 and EDC signaling (*Li et al., 2007*). Indeed, since a critical step in steroid hormonal action is the dimerization of ligand-bound receptor monomers, computational modeling showed homodimerization of ligand-bound receptors to be an inherently nonlinear process (Fig. 2). Furthermore, heterodimerizaton ($L_{en}ER-ERL_{ex}$; ER receptors bound to both endogenous- $L_{en}$ and exogenous ligands- $L_{ex}$) can also induce U-shaped responses with the magnitude of response varying with the transcriptional activity of the heterodimer. The model shows that a U-shaped response arises when the exogenous ligand $L_{ex}$ is an agonist, regardless of the heterodimer being absent, or being pure/partial activator or repressor. On the other side, an inverted U-shaped response appears when the heterodimer is a pure or partial activator, regardless of whether the exogenous ligand is an agonist or antagonist. Monotonic response, however, only occurs in two simulated scenarios i.e., with the treatment with antagonist when mixed-ligand heterodimer is a repressor or absent.

## Role of estrogens in gestational insulin resistance

During pregnancy, gestational insulin resistance develops naturally to direct nutrients to the embryo; however, the plasma glucose levels may rise to promote gestational diabetes. Insulin resistance perturbs the glucose metabolism and manifests itself in attenuated glucose uptake in the skeletal muscle, white adipose tissue and liver and inadequate suppression of glucose production in the liver (*Catalano et al., 2003*). The body adapts to this disturbance by increasing the biosynthesis and secretion of insulin through beta-cells located in the pancreatic islets (islets of Langerhans) and by increasing their cell mass. In the case of GDM, beta-cells do not respond adequately to the changes in the organism which leads to elevated blood glucose levels. It is possible that in case of GDM, beta cells fail to respond adequately to the natural hormonal signaling during pregnancy, particularly to estrogens (estradiol, estrone and estriol) and progesterone during the second trimester, when both GDM and increase in estrogens/progesterone concentrations might co-occur (10-fold increase in progesterone, 30-fold increase in estrogens) (*Nadal et al., 2009b*). In addition, both estrogen and progesterone receptors are expressed in the pancreatic beta-cells (*Nadal et al., 2009a*) indicating their responsiveness to the pregnancy hormones.

Three major mechanisms that are involved in counteracting insulin resistance during pregnancy are increasing the biosynthesis of insulin, enhancing the sensitivity of pathways for glucose stimulated insulin secretion, and hypertrophy of the beta-cells. Indeed, E2 has been shown to increase glucose stimulated insulin secretion (*Nadal et al., 1998*). E2 interacts with a membrane-associated receptor and closes ATP-dependent $K^+$ channels. Subsequently, this depolarizes the plasma membrane and allows the influx of $Ca^{2+}$, which in turn triggers the release of insulin. When E2 was applied to isolated mouse islets of

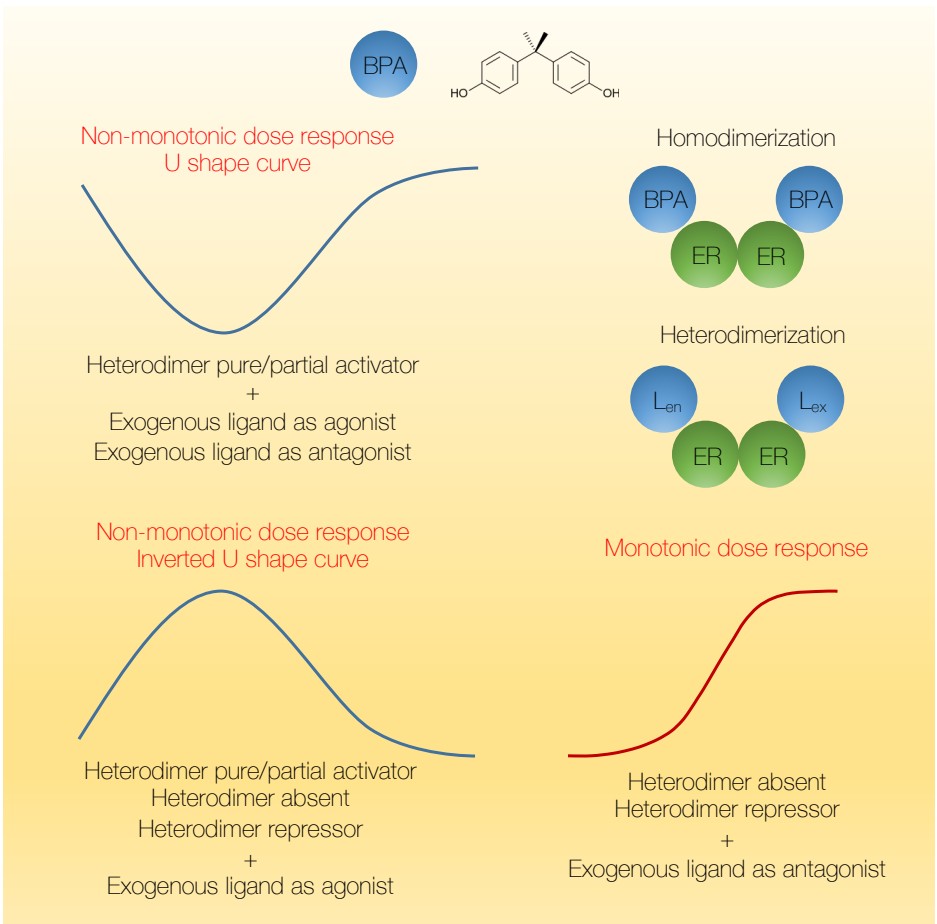

**Figure 2** **The prediction of non-monotonic dose response to BPA.** BPA may induce non-monotonic dose response as manifested by the appearance of the inverted U-shape curve, e.g., when insulin content is measured in isolated pancreatic islets after treatment with increasing concentrations of BPA (0.1–1,000 nM) for 48 h (*Alonso-Magdalena et al., 2008*). Computational modeling that takes into account the dimerization kinetics of the estrogen receptors and binding to either endogenous or exogenous ligand, or both (heterodimerization) accurately predicts the appearance of non-monotonic dose responses (both the U-shape and inverted U-shape curves). U-shaped dose response appears in case of the heterodimer $L_{en}$ER-ERL$_{ex}$ (also named LXXR) acting as a pure or partial activator, regardless of the nature of exogenous ligand. Non-monotonic inverted U-shape curve appears when exogenous ligand is an agonist, regardless of the activity of the heterodimer. Finally, monotonic response can also arise in two cases. $L_{en}$, endogenous ligand; $L_{ex}$, exogenous ligand.

Langerhans in the absence of glucose stimulation it produced no effect, while in the presence of 8.3 mM glucose an increase in oscillatory electrical activity followed with an oscillatory pattern of $Ca^{2+}$ concentration were observed, indicating glucose dependent E2 effect. This pattern of $Ca^{2+}$ oscillations induced pulsatile insulin secretion (*Martin, Sanchez-Andres & Soria, 1995*) and a total increase in insulin secretion of 30%. Therefore, E2 exerts its effect on pancreatic beta cells that parallels those observed during normal pregnancy. A wide range of physiological E2 concentrations (100 pM–1M) were shown to increase intracellular calcium concentrations; however, even the suboptimal pM concentrations (10 pM) were

able to show a slight effect. The same effect was observed *in vivo* (*Alonso-Magdalena et al., 2006*), when adult male mice were injected twice a day with E2 or a vehicle for four days. Insulin content measured with immunocytochemistry showed that beta-cells from E2 mice had increased staining compared to the vehicle group and therefore possessed higher insulin content in each individual cell that was measured (similar effect was observed for the BPA treatment).

In addition, it has been shown that stimulation of islets isolated from adult mice with E2 at physiological concentrations increases insulin content through promoting insulin mRNA synthesis (*Alonso-Magdalena et al., 2008*). Insulin mRNA levels increased 1.6 fold only 6 h after treatment of the islets with E2 (10 nM). In the long-term stimulation (48 h) E2 showed similar increase in total insulin content, which fully paralleled the effect of BPA at concentrations of 1 nM. BPA titration displayed inverted U-shape dose-response curve indicating the existence of an active concentration range. A similar narrow concentration window of physiological activity was observed for E2, as islets treated with the increasing doses of E2 for 48 h showed non-monotonic dose response (inverted U-shape) with the maximal response at E2 concentrations 1–10 nM. Consistently, the treatment of pancreatic beta-cell line MIN6 for 2 h with various concentrations of E2 produced an inverted U-shaped curve (maximum of insulin secretion at E2 concentrations 1-10nM) (*Sharma & Prossnitz, 2011*). This indicates that stabilizing effect on gestational insulin resistance through increasing insulin biosynthesis and secretion in the islets may be highly regulated biological process directed through a narrow range of hormonal concentrations.

Glucose homeostasis is primarily controlled at the level of glucose uptake by the skeletal muscle (SM) and white adipose tissues (WAT). Estrogen receptors ERα and ERβ are expressed in SM and WAT and have opposing effects on expression of GLUT4 glucose transporter. GLUT4 is the main glucose transporter in SM and WAT, and it accelerates the transport of glucose under insulin stimulation (*Barnard & Youngren, 1992*; *Barros et al., 2009*). With insulin binding to its receptors and activation of the tyrosine kinase signaling pathway, GLUT-4 receptors are being transported from the intracellular depos to the sarcolemmal membrane (SL) in SM or plasma membrane in WAT. With GLUT4 being deposited in the membrane, glucose can diffuse into the cell (facilitated diffusion). GLUT4 defective synthesis, glycosylation, translocation and anchorage to the membrane could promote development of insulin resistance. The effect of estrogen on insulin resistance at the level of peripheral tissues is mediated by its two receptors, ERα and ERβ, and sometimes perplexing and contradictory role of estrogen in IR might be explained by the opposing functions of these receptors. ERβ knockout mice show signs of hypoglycemia, islet hypertrophy and elevated plasma insulin after glucose stimulation, while ERα knock-out mice show completely opposite phenotypes of hyperglycemia and glucose intolerance. In addition, expression of GLUT4 in ERβ knockout mice is increased, while ERα knockout mice possess lower GLUT4 expression (*Barros et al., 2006*; *Barros, Machado & Gustafsson, 2006*). This indicated that ER β activation could have a diabetogenic effect, while ERα action is important for the maintenance of normal glucose homeostasis. Therefore, in case that ERα and ERβ are exhibiting diametrically opposed effects, cells and tissues with both receptors expressed would respond to E2 in opposite ways depending primarily on the

ERα to ERβ ratio (net effect). In addition, it has been shown that ERα is dominating in the white adipose tissue, while ERβ is more distributed in the skeletal muscle, where E2 as a net effect might promote insulin resistance (*Barros et al., 2009*).

The role of estrogens in developing insulin resistance during pregnancy, a feature that chronic bisphenol A exposure is predicted to be mimicking, is somewhat dichotomous, as some studies show protective effect of estrogen on insulin resistance. In a 2012 study, male and ovariectomized female C57BL/6J mice had higher propensity to developing insulin resistance compared to non-ovariectomized females when exposed to a high-fat diet, while the administration of E2 to ovariectomized females reduced insulin resistance in both high and low fat diet groups, as measured by area under curve (AUC) in the glucose tolerance test (*Stubbins et al., 2012*). The study did not show the effect of E2 on non-ovariectomized females, therefore drawing conclusions only on the stabilizing effect of estrogen on hormone-deficient mice. Similarly, ArKO mice (transgenic mice with inactivated aromatase enzyme, essential for E2 synthesis) developed glucose intolerance and insulin resistance, an effect reversible by E2 (*Barros & Gustafsson, 2011*). Likewise, ERα -/- mice are glucose intolerant and insulin resistant (*Heine et al., 2000*). A study in rats found that treatment of male rats with E2 protected against accumulation of fatty acids in pancreatic islets and against pancreatic beta cell failure (*Tiano et al., 2011*), therefore preparing the islets for increased insulin production during the pregnancy and gestational insulin resistance. Concurrently, the study proposes ERα or ERβ receptors as promising therapeutics to prevent beta cell failure in T2D. Estrogens, although with a protective role on beta-pancreatic cells through ER receptors signaling, in peripheral tissues like SM were predicted to promote insulin resistance via ERβ net effect (ratio ERβ/ERα) (*Barros et al., 2009*). However, some studies showed that E2 treatment might result only in slight decrease in glucose uptake in SM without insulin and with significant improvement of muscle glucose uptake with insulin treatment, which suggests ERα dominance over ERβ in SM (*Gorres et al., 2011*; *Inada et al., 2016*), leading to contention within the field.

On the other side, insulin resistance and hyperinsulinemia are common among women consuming oral contraceptives containing estrogens and during pregnancy, and diminished peripheral glucose uptake was observed among normal subjects treated with ethinyl estradiol (*Polderman et al., 1994*). Association has been found between insulin resistance and E2 levels in obese children (*Lin & Ji, 2016*). Studies suggest that in humans the lipolytic effect of placental lactogen directs maternal metabolism toward lipids, rather than glucose utilization, in the same time adding to the preservation of glucose for the fetus (*Baz, Riveline & Gautier, 2016*). In non-diabetic women, employing the intravenous glucose tolerance test in 296 oral contraceptive users and 95 nonusers, estrogen based contraceptives reduced the glucose elimination constant and reduced insulin sensitivity by 30–40% (*Godsland et al., 1992*). Other studies showed that users of oral contraceptives with synthetic estrogen had up to 61% higher plasma glucose levels, up to 40% higher insulin response and up to 40% higher C peptide response, a method to determine residual insulin secretion (*Godsland et al., 1990*). More recent meta-analysis showed less disturbance in carbohydrate metabolism (*Lopez, Grimes & Schulz, 2014*), potentially due to the change in composition of contraceptives that contain lower estrogen content.

## Strong cross-generational transmission of BPA effects to male offspring

In 2010 came the first study in mice that showed that treatment of pregnant females with low doses of BPA could in F1 male offspring provoke glucose intolerance, insulin resistance and altered beta-cell function (*Alonso-Magdalena et al., 2010*). The pancreatic islets from male offspring showed abnormal $Ca^{2+}$ signaling and increased insulin secretion. Decreased bromodeoxyuridine (BrdU) incorporation into insulin-producing cells was detected in the male offspring, indicating absence of proliferation. An indication of non-monotonic dose response was observed as insulin sensitivity was evidently impaired in F1-BPA10 male offspring (slight alteration in 3-months old mice, significant alteration in 6-months old mice), however no significant alteration of insulin sensitivity was detected in F1-BPA100 males. Similarly, both serum insulin levels and glucose-stimulated insulin secretion *in vivo* and ex vivo (using isolated islets) were higher in F1-BPA10 males compared to the control, while the same effect was not observed in F1-BPA100 male mice, again suggesting non-monotonic dose response. Similar non-monotonic dose response was observed in the increase of intracellular $Ca^{2+}$ in isolated islets after glucose stimulation (both 7mM and 16mM glucose) that was detected in the F1-BPA10, but not F1-BPA100 males. In addition, F1-BPA10 male mice weighted 3% more than the control mice, while F1-BPA100 were 4.5% lighter at birth than controls. On the other side, the glucose tolerance was altered in both F1-BPA10 and F1-BPA100 male mice, however with a slight decrease in AUC in F1-BPA100 males. The phenotype of altered glucose metabolism in male offspring was confirmed subsequently in additional studies (*Wei et al., 2011*; *Angle et al., 2013*; *García-Arevalo et al., 2014*; *García-Arévalo et al., 2016*).

In addition, the interaction of two environmental stimuli, BPA and high fat diet (HFD) has been examined (*García-Arevalo et al., 2014*). The group of F1 male mice exposed to BPA in utero increased their weight starting at 18 weeks old and reached the HFD and HFD-BPA groups at week 28. The BPA group showed similarities to both HFD and HFD-BPA groups in exhibiting fasting hyperglycemia and glucose intolerance (slight increase in AUC at 17 weeks, and reaching HFD and HFD-BPA groups at 28 weeks). Other studies have found similar tendencies of increased body weight, elevated serum insulin levels and impaired glucose tolerance in F1-BPA offspring, which were elevated and accelerated with the introduction of a high fat diet (*Wei et al., 2011*). Consistent with the BPA-HFD interaction, F1-HFD-BPA mice showed severe metabolic syndrome, including obesity, dyslipidemia, hyperglycemia, hyperinsulinemia, and glucose intolerance. However, the effects were present on lower BPA doses, but were absent on higher BPA doses indicating that, even though it triggered adverse metabolic effects of BPA, HFD did not influence the non-monotonic dose response to BPA. Another study showed that the fetal and perinatal treatment with BPA was more detrimental than neonatal (*Liu et al., 2013*). The effect of BPA on blood glucose homeostasis in six months old male mice was pronounced in the group with fetal exposure from day 6 of pregnancy until postnatal day (PND) 0, while only modest effects were observed in the group with neonatal exposure (PND0–PND21), suggesting a critical timeframe when BPA can exert its effects during the embryological development.

Furthermore, a plethora of diverse effects on male offspring were detected (including a decrease in glucose tolerance and serum adiponectin and an increase in body and liver mass, abdominal adipocyte number, and serum leptin and insulin levels). This occurred at the doses of BPA below the predicted 'no adverse effect level' (NOAEL), while doses 10-fold above NOAEL did not show any significant effect, confirming the non-monotonic dose response (*Angle et al., 2013*). In addition, the BPA exposure might predispose offspring to the fatty liver disease, a part of the metabolic syndrome symptoms (*Jiang et al., 2014*). In Wistar rats with perinatal exposure to BPA, at 26 weeks of age extensive fat accumulation in liver was observed, as well as elevated serum alanine aminotransferase (ALT), the indicator of liver damage. Milder effects were observed at three weeks, i.e., a decrease in mitochondrial respiratory complex (MRCI, MRCIII) activity and altered expression levels of genes involved in mitochondrial fatty acid metabolism, while at 15 weeks an infiltration of liver cells with fat (steatosis), as well as upregulation of lipogenesis genes and increased levels of reactive oxygen species (ROS, indicative of the loss of mitochondrial function) were observed.

It has been shown, using different animal models, that the prenatal exposure to BPA leads to severe glucose intolerance, insulin resistance and hyperinsulinemia during postnatal life (*Alonso-Magdalena et al., 2015*). In mice, BPA-treated pregnant females (10 µg/kg on days 9–16 of gestation) produced male progeny that showed altered glucose metabolism at 17 and 28 weeks of age (*García-Arevalo et al., 2014*), therefore confirming that the treatment with endocrine-disrupting chemicals leads to the perturbation of glucose metabolism of pregnant females that is being efficiently transmitted to the offspring. Recently, it has been shown that BPA treatment of pregnant female mice (10 and 100 µg/kg per day) in male offspring promotes increased expression of cell division genes in the beta cells of pancreas followed with increased pancreatic beta-cell growth and increased insulin levels at postnatal days 0, 21 and 30 (*García-Arévalo et al., 2016*). Conversely, at postnatal day 120 beta cell mass diminished and mice showed increased fasting glucose levels and tendency towards glucose intolerance. Therefore, parental BPA exposure in mice leads to the surplus of insulin signaling during early life in male offspring that could advance into the impaired glucose tolerance of adulthood. The perturbations in glucose metabolism induced by EDC could therefore be actively transmitted to the developing mouse embryo and appear as long-term consequences later in life (Fig. 3). A 2016 study in humans showed association of prenatal creatinine-adjusted urinary BPA concentrations with BMI levels and waist circumference in male children of 1–4 years of age (*Vafeiadi et al., 2016*). For female offspring, prenatal urinary BPA was inversely associated with BMI and adiposity measures, confirming similar gender-related trends that were observed in animal studies.

## BPA effects detected in female offspring in mice

Interestingly, even though the effects of BPA exposure in mice are usually observed in the male offspring, multiple studies have reported the occurrence of strong effects in the female offspring. In one study, female offspring showed altered blood glucose homeostasis, as measured with intraperitoneal glucose tolerance test (ipGTT) (*Liu et al., 2013*). At three months, the response to glucose challenge was evident in the group with fetal BPA

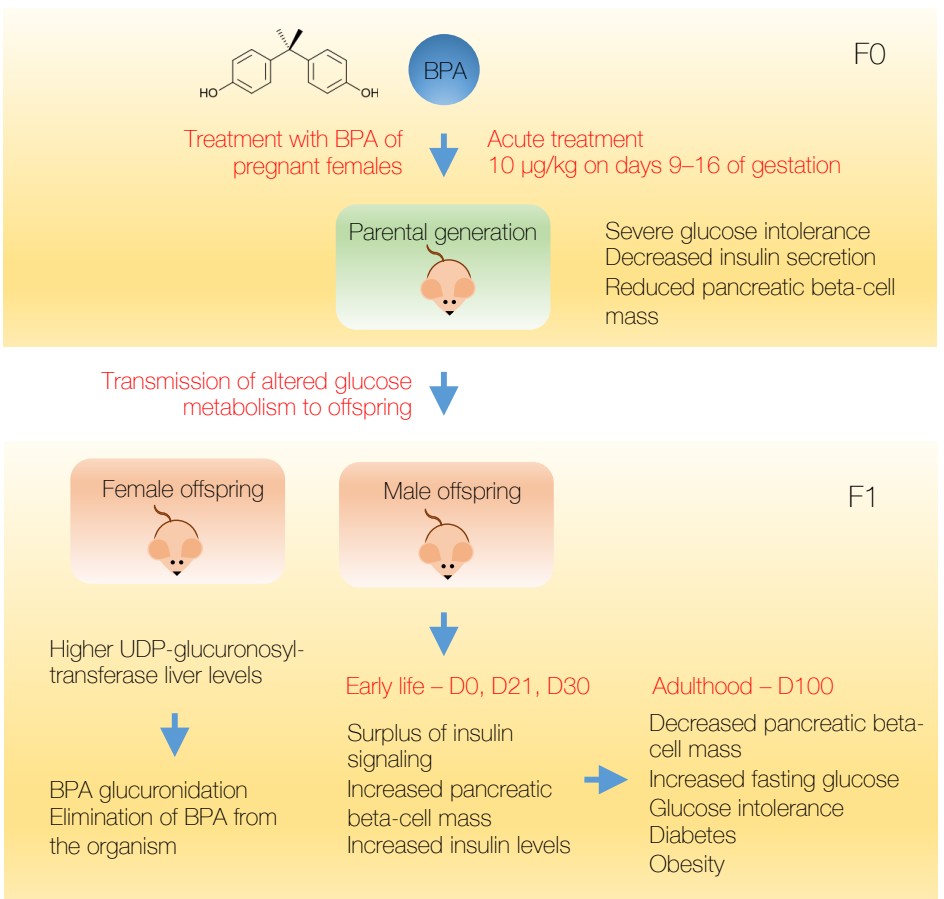

**Figure 3** **Effective transmission of BPA effects to male offspring in mice.** Acute BPA treatment during gestation leads to severe glucose intolerance, decreased insulin production, and altered glucose metabolism that is being transferred to the male offspring. During the early life in male offspring there is a surplus in insulin signaling and insulin production that ultimately leads to decreased pancreatic beta mass and glucose intolerance in adulthood. Female offspring is protected from the BPA effects due to the higher levels of the enzyme involved in BPA glucuronidation process and elimination of BPA from the organism.

exposure from day 6 of pregnancy until PND 0 (P6–PND0) compared to the groups with preimplantation (P1–P6), neonatal (PND0–PND26) or fetal plus neonatal (P6–PND26) exposure, indicating the existence of the critical window of exposure similar to the effect in male offspring. At six months, the effects are significant for both P1-P6 and P6–PND0 groups, however the intensity of alterations in glucose homeostasis decreases. Finally, at eight months, AUC in ipGTT in all groups did not differ from the control group of mice. In addition, P6-PND0 female offspring at three months showed increasing fasting insulin levels, while at six months they showed increased fasting glucose levels. The insulinogenic index ($\Delta I_{0-30}/\Delta G_{0-30}$), which is measuring insulin release in response to the glucose load, was decreased in three months old females from the P6–PND0 group, and then improved in six and eight months old females. A study in rats showed similar profiles between male and female offspring rats in fasting blood glucose and fasting serum insulin at three, nine, 15, and 26 weeks after perinatal exposure to 50 µg/kg per day of BPA (*Wei et al., 2011*).

The group of HFD-BPA female rats showed higher fasting blood glucose and fasting serum insulin levels starting from nine weeks of age compared to the HFD group, indicating the interaction of BPA exposure with the HFD phenotype. In addition, at 26 weeks, female BPA-exposed offspring on a normal diet, similar to the male offspring, had greater glucose and insulin AUC in OGTT and ITT compared to controls. Concordantly, the interaction of HFD phenotype and BPA exposure could be observed in females at 26 weeks in OGTT AUC.

In a 2017 study, fetal livers were collected from the F1 generation (embryonic day E18.5) exposed to BPA in utero from E7.5 to E18.5, and gene expression of key hepatocyte maturation markers was assessed (*DeBenedictis, Guan & Yang, 2016*). In particular, female mice changes were prominent and involved decreasing levels of mature hepatocyte markers, such as albumin and glycogen synthase (reduced 65% and 40%, respectively), decreased levels of C/EBP-alpha, the master transcription factor of hepatocyte maturation (reduced 50%), and increased levels of immature hepatocyte marker, α-fetoprotein (increased 43%). In addition, the markers of proliferation (PCNA) were elevated (40% increase) and the markers of apoptosis (caspase-3) were decreased (40% decrease). This suggests that BPA reduces maturation and alters the balance of proliferation and apoptosis in fetal hepatocytes, in a similar direction as observed in the pancreatic islets in F1 mice (*García-Arévalo et al., 2016*), where upregulated genes enrich in gene ontologies related with cell cycle, mitosis and cell division.

## Epigenetic modifications of DNA as a mechanism of BPA-induced transgenerational effects

In another study, it has been shown that BPA might promote epigenetic changes that are associated with the transmission of perturbed carbohydrate metabolism phenotype to offspring (*Susiarjo et al., 2015*). Exposure of BPA in C57BL/6 mothers produced multigenerational metabolic abnormalities and stable inheritance of changes in DNA methylation at the imprinted *Igf2* gene locus, a well-studied gene locus involved in fetal growth and recently shown to alter adult energy metabolism, fat deposition and obesity (*Jones, Levorse & Tilghman, 2001*). Study established altered body mass, glucose tolerance, and insulin secretion in male, but not female, F1 mice and the effect on F2 generation was estimated to originate from the exposure to BPA in developing germ cells. Previous study has shown that BPA can alter the methylation levels of differentially methylated regions (DMR) and that it promotes loss of monoallelic expression of the *Snrpn, Igf2 and Kcnq1ot1* genes in a tissue specific manner in F1 offspring (placental and embryonic tissue) (*Susiarjo et al., 2013*). Loss of imprinting was highly specific for placenta as 13/28 placentas showed loss of imprinting (LOI) compared to 0/23 controls for the *Snrpn* gene. On the contrary, at the *Igf2* gene locus, LOI was found in 7/28 embryos compared to 0/23 controls, while it has been absent in placentas. Repressed maternal *Igf2* allele was shown to increase its expression up to 68.9% of the total *Igf2* expression. Analysis of DNA methylation at the 16 CpG sites in the *Snrpn* promoter, which is hypermethylated on the maternal allele, showed reduction in methylation levels in upper dose BPA exposed placentas (observed methylation levels of the maternal allele were 53.8% lower than those of controls), while

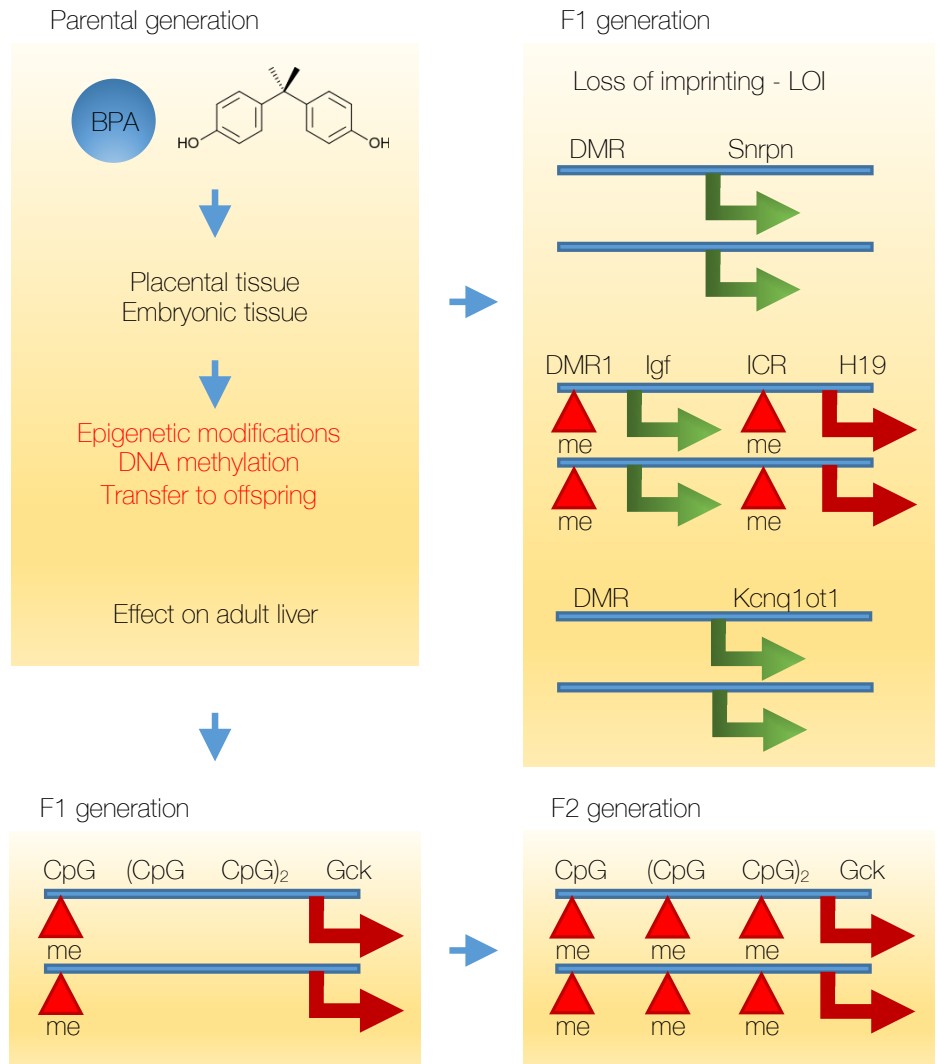

**Figure 4  BPA induced epigenetic modifications are actively transmitted to offspring.** BPA can induce changes in DNA methylation in the placental tissue, as well as in the embryo of the F1 generation. Modifications in DNA methylation at the DMR—differentially methylated regions induce changes in gene expression and LOI—loss of imprinting (loss of monoallelic expression) of at least three imprinted genes (*Snrpn*, *Igf2* and *Kcnq1ot1*). In the case of *Igf2* increased methylation leads to the loss of expression of the neighboring *H19* gene from the maternal allele and subsequent biallelic expression of *Igf2*. In the case of *Snrpn* and *Kcnq1ot1* the loss of methylation at the DMR leads to biallelic expression of these genes. In the liver in F1 and F2 generations, DNA methylation of the *Gck* (glucokinase) gene promoter decreases its expression. In F1 generation *Gck* contains hypermethylated one CpG island (out of 5 CpG islands in total present in the promoter of the *Gck* gene). In F2 generation, DNA methylation is increased as all 5 CpG islands become methylated.

a similar trend was observed in the embryos. On the Igf2 locus, the DMR1 region showed significantly increased methylation levels in embryos (45.6% in controls vs. 55.7% in upper dose BPA in embryos, $p < 0.05$; trend not observed in placentas), which was consistent with previous association of the gain of DNA methylation with biallelic *Igf2* expression (Fig. 4). To determine whether genomic imprinting effects were present in the F2 offspring

in the absence of further exposure, F1 females were mated to unexposed males; however, biallelic expression of the *Igf2* gene was not observed (*Susiarjo et al., 2015*). Nevertheless, BPA treated embryos in F2 generation showed significant overexpression of *Igf2* gene compared to controls. Similar to the BPA phenotype, $H19^{\Delta 3.8/+}$ male mice with the imprinted control locus (ICR) deletion showed increased glucose intolerance, increased body weight and body fat content, indicating that BPA might exert its effect partly through epigenetic changes at the *Igf2* locus.

Similarly, another study has shown that BPA can exert multigenerational effects, as the F2 generation rats were found to exhibit glucose intolerance and insulin resistance in ipGTT and ipITT and decreased expression of *Gck* (glucokinase) gene in the liver, and this effect was estimated to be epigenetic. The *Gck* promoter in the hepatic tissue in F2 generation rats exhibited fully methylated status in the all five CpG sites upstream of the promoter (located up to 314 bp upstream), compared to the unmethylated status in the control rats (*Li et al., 2014*). A similar trend of BPA-induced *Gck* promoter hypermethylation was observed in F1 generation in three week old offspring rats and more pronounced increase of *Gck* methylation was observed in 21 week old rats (*Ma et al., 2013*). In addition, in both studies, BPA was shown to induce changes in global DNA methylation (decrease in DNA methylation, 84.3% BPA-treated F2 offspring vs 90.1% controls) implicating long-term epigenetic effects.

## Epidemiological studies connecting BPA and T2D

Multiple epidemiological studies have linked BPA exposure (mainly measured as urinary BPA concentrations) with T2D or prediabetes occurrence (diagnosed using lab tests of fasting blood glucose or HbA1c, self-reports, previous doctor diagnosis, or current use of diabetic drugs) (*Sowlat et al., 2016*). A study of 3,516 subjects from the National Health and Nutritional Examination Survey (NHANES) 2003–2008 found a positive correlation between higher urinary BPA and prediabetes (fasting plasma glucose), and correlation was independent of confounders including body mass index, alcohol intake, blood pressure and serum cholesterol (*Sabanayagam, Teppala & Shankar, 2013*). When T2D was examined within the participants of NHANES, a correlation was observed with odds ratio (OR) of 1.39; however, only for the cycle 2003–2004 (*Lang et al., 2008*; *Silver et al., 2011*, pp. 2003–2008). Another study of NHANES participants found multivariable-adjusted OR of 1.68 when comparing 1st quartile of urinary BPA concentrations with 4th quartile ($p = 0.002$) (*Shankar & Teppala, 2011*). In addition to measured urinary BPA concentrations, a study of 2,581 subjects in Thailand that involved blood (serum) BPA levels has found a significant correlation between serum BPA levels and T2D (OR = 1.83 and 1.88, for women and men, respectively) (*Aekplakorn, Chailurkit & Ongphiphadhanakul, 2015*). However, other epidemiological studies failed to find a statistically significant correlation of BPA levels and T2D. For example, a study involving 3,423 subjects in China measured urinary BPA concentrations and found slightly increased OR for T2D (OR = 1.30, 1.37, in second and fourth BPA quartile, respectively), but not third quartile, and the overall association was not significant (*Ning et al., 2011*). Similarly, a cross-sectional study of Korean adults showed increased adjusted odds ratio for T2D in the upper BPA quartile (OR = 1.89); however,

this association did not reach statistical significance (*Kim & Park, 2013*). Therefore, the epidemiological evidence for the connection between human BPA concentrations and T2D requires further assessment.

## BPA effects in promoting cardiovascular diseases

The previous studies on BPA treatment in animal-models found evidence of interference on the mechanisms underlying insulin signaling and diabetes, while the underlying mechanisms of association with cardiovascular diseases are not evident. In a 2008 study (*Lang et al., 2008*) 1,455 adults from the NHANES 2003–2004 study (694 men and 761 women), 18–74 years of age, have had their urinary BPA and creatinine levels measured. Regression association was adjusted for the creatinine concentration in urine, as well for a set of standard factors, such as age, sex, ethnicity, education, and body mass index. Tested samples provided 80% power and detected that higher BPA concentrations in urine were associated with cardiovascular diagnoses in models adjusted for age and sex and in a fully adjusted model (OR = 1.39 per 1-SD increase in BPA, 95% confidence interval: 1.18–1.63, full adjustment $p$val = 0.001). Similarly, in the study of 1,493 participants of the NHANES 2005–2006 elevated urinary BPA concentrations were correlated with coronary artery disease (CAD) (OR = 1.33) (*Melzer et al., 2010*, p. 6). In mice, BPA exposure has been linked to the increase in production of a strong helper T type 1 (Th-1) type cytokine (IFN-gamma) in lymphocytes (*Youn et al., 2002*), which given the role of Th-1 immunity and inflammation in atherosclerosis and CAD (*Engelbertsen & Lichtman, in press*; *Pjanic et al., 2016*; *Kim et al., 2017*) might mechanistically bridge the link to CAD. Whether BPA exerts its effect on cardiovascular diseases through its loose binding to the estrogen receptor or via binding to the estrogen related receptors, and whether this is a shared downstream mechanism with the effect on insulin resistance remains to determined, especially considering that estrogen receptor signaling exhibits pleiotropic effects on the cardiovascular system.

## Proposed BPA involvement in other phenotypes

In epidemiological studies, bisphenol A exposure has been linked to various disorders in humans, such as insulin resistance and diabetes (*Shankar & Teppala, 2011*), cardiovascular diseases (*Lang et al., 2008*), and obesity (*Wang et al., 2012*). In a 2008 study (*Lang et al., 2008*), higher BPA concentrations were associated with diabetes mellitus (OR = 1.39 per 1-SD increase in BPA, 95% confidence interval: 1.21–1.60, full adjustment $p$val < 0.001). In addition, in the same study, out of eight blood serum analytes, urinary BPA was associated with clinically abnormal concentrations of the liver enzymes γ-glutamyltransferase (OR = 1.29 per 1-SD increase in BPA, 95% CI [1.14–1.46], full adjustment $p$val < 0.001), alkaline phosphatase (OR = 1.48, 95% CI [1.18–1.85], $p$val = 0.002) and lactate dehydrogenase ($p$val = 0.04). As no significant associations with the other common disorders were found, the specificity of the associations to insulin resistance, diabetes and cardiovascular diseases implicated BPA in modulation of common mechanisms perturbed in these diseases. On the other side, the association of BPA and the enzymes present in liver, specifically γ-glutamyltransferase and lactate dehydrogenase, was preserved in a cohort

without cardiovascular diseases or diabetes (glutamyltransferase OR = 1.22, 95% CI [1.02–1.45], $p$val = 0.03; lactate dehydrogenase OR = 1.31, 95% CI [1.06–1.62], $p$val = 0.01) (*Lang et al., 2008*), suggesting that mechanisms underlying the BPA effect on liver are distinct from the cardiovascular and insulin resistance/diabetes effects and therefore exclude reverse causation of these diseases. In addition, in patients with BMI less than 25, BPA preserved significant association with γ-glutamyltransferase ($p$val = 0.03).

## DISCUSSION

In the last two decades, bisphenol A has been a target of strong public and scientific scrutiny. The number of papers on BPA available on PubMed reaches 10,668, with several hundred papers published each year. An overwhelming body of knowledge has accumulated since, both mechanistic, in animal models, and epidemiological, that has contributed to our better understanding of the implications that the widespread and chronic exposure of human populations to BPA carries. Even though BPA properties as an estrogen mimicking molecule have been discovered in 1936 (*Dodds & Lawson, 1936*; *Dodds & Lawson, 1938*; *Dodds & Lawson, 1938*), its widespread use as a synthetic polymer unit, starting from the late 1950s, hasn't been influenced by the fact that it might behave as an endocrine-disrupting chemical. Driven by the industrial tendencies and novel emerging markets, BPA-based polycarbonate polymers have infiltrated almost every aspect of the human life, including food containers, baby and water bottles, can and glass linings, various medical and dental devices, eyeglass lenses, and finally the epoxy lining of water pipes and tanks, making the large majority of human populations chronically exposed to the low levels of this synthetic chemical.

Whether the widespread use of BPA in the contemporary human environment is related to the expansion of insulin resistance, diabetes and obesity-related diseases is unclear. One can contemplate that this is most probably not a direct or unique cause of the elevated fasting plasma glucose levels, insulin resistance and diabetes expansion in the human populations, however the time frames of the diabetes expansion and the use of plastic bottles coincide, hence, given the experimental findings, the question becomes more quantitative than qualitative. The prevalence of hyperglycemia and diabetes are rising globally since 1980 with a mean fasting plasma glucose level increasing 0.09 mmol/L per decade, while the number of people with diabetes increased from 153 million in 1980 to 347 million in 2008, more than doubling in size during three decades (*Danaei et al., 2011*). The level of BPA exposure in human populations depends primarily on how chemically effective the hydrolysis and photo-degradation of polycarbonate polymers are in their natural environment and that depends on the content, stability and storage conditions of plastic polymers and coating materials. On higher temperatures, increased hydrolysis leads to the excess of leached BPA in the neighboring environment. Certain polycarbonate plastics and coatings may represent common sources of leaching BPA levels, e.g., food containers that will be exposed to higher temperatures in microwave ovens. Therefore, unknowingly humans may further increase the hydrolysis of polycarbonates and subsequently their exposure to BPA by, e.g., microwaving food in plastic containers, refilling plastic water
bottles or leaving plastic water bottles in the sun exposed to light, with BPA polymers undergoing photo-oxidative degradation.

BPA traces have been detected leaching from the polycarbonate plastic products, as well as present in various human tissues. BPA environmental levels correspond to the tissue levels, appearing in the concentrations of the same order of magnitude (ng ml$^{-1}$), indicating effective transfer from the environment to the human internal organs and tissues. BPA has been detected in human serum with concentrations up to 4.4 ng ml$^{-1}$ (*Vandenberg et al., 2007*) and urine with detection rates up to 97.5% (*Yang et al., 2006*). As tissues that exhibited the highest BPA concentrations up to the levels of 11.2 ng ml$^{-1}$ were related to the embryo development, such as placenta, umbilical cord, and amniotic fluid, as well as to the maternal influence of postnatal development of infants, like breast milk and colostrum, it may not be surprising that experimental studies in mice, as well as epidemiological studies in humans, showed pronounced transgenerational effects of BPA. The question of mechanism for gender-related differences of the BPA effects in offspring still remains open, as to why predominantly male offspring exhibit increased insulin resistance, while female offspring show negative BMI correlation. The explanation may come from a gender-related differences in BPA-processing liver enzyme levels and subsequent BPA clearance from the organism. For instance, it has been shown that female rats harbor higher UDP-glucuronosyltransferase liver levels, as well more effective BPA glucuronidation reaction that eliminates BPA from the organism (*Takeuchi et al., 2004*).

Skepticism could emerge due to the fact that BPA has much lower affinity for estrogen receptors, therefore questioning whether its effects are indeed negligible. However, it may be possible that effects of prolonged exposure to low affinity binders mimic the short term effect of high affinity binders, providing mechanistic explanation for direct BPA action. Indeed, low affinity binders may have profound effects on the pathogenesis of obesity and insulin resistance, as shown in the case of insulin-like growth factor binding proteins (IGFBPs) comprising both insulin-like growth factor (IGF) high- and low-affinity binders (*Kim et al., 1997*; *Ruan & Lai, 2010*).

As it has been shown that in various *in vivo* and *in vitro* experiments BPA effects follow the non-monotonic dose response (i.e., the inverted U-shape curve), a narrow concentration range might exist that is critical for the BPA action. Similarly, the narrow window of developmental stages exists during which BPA will exert its maximal effect (e.g., P6–PND0). Combined with the gender related differences, there might exist a specific set of conditions under which BPA exerts its maximal biological and physiological effect (Fig. 5). Therefore, the complete elucidation of the maximal BPA effects on offspring may be limited to the very specific set of experimental conditions. In addition, the discovery of BPA mechanism that involves epigenetic modifications (i.e., DNA methylation) reveals the complexity of the mechanism responsible for the appearance of BPA phenotype in subsequent generations.

Even though at present day BPA-free plastic products are getting more available, e.g., BPA-free water bottles, the use of BPA-free polymers has not reached widespread levels and in many cases BPA is simply substituted with one of its analogues, BPS or BPF, that may exhibit similar behavior to BPA. In addition, the level of public education on this subject
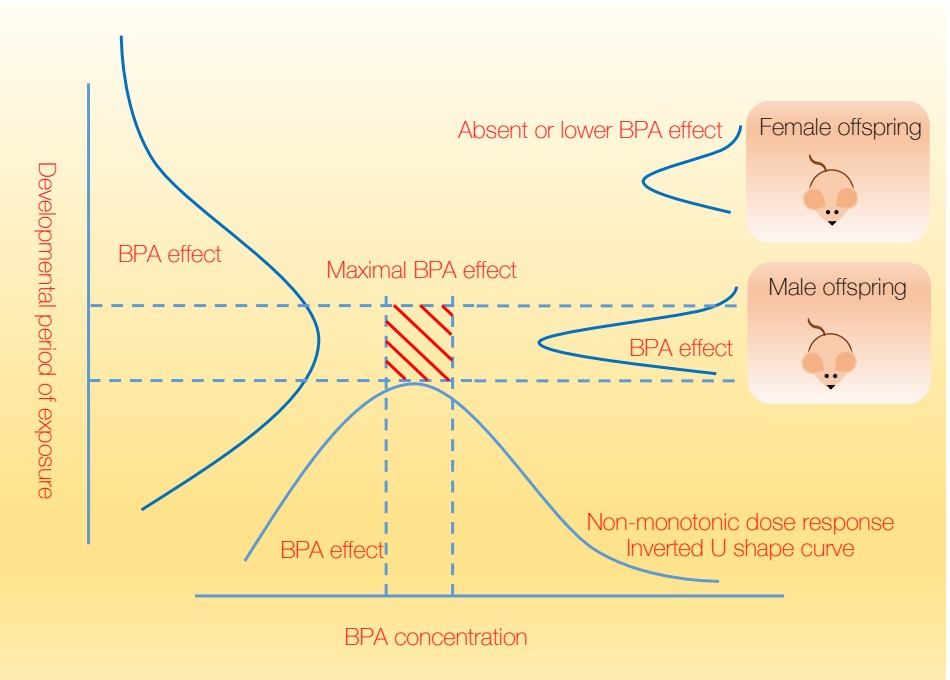

**Figure 5** **Maximal biological effect of BPA is confined to a narrow range and dependent on dose, gender and developmental stage.** BPA exerts effects that follow non-monotonic dose response (e.g., the inverted U-shape curve), therefore a narrow concentration window might exist that is essential for the BPA action. Concordantly, BPA will show its maximal effect in the narrow window of developmental stages (e.g., P6–PND0). These confined ranges of maximal BPA effects, together with the gender related differences, comprise a specific set of conditions for the maximal biological and physiological effect of BPA.

remains relatively poor and the amount of BPA present in the environment still remains at the levels of a substantial health threat. Consistently, it will take years of regulation of environmental and industrial BPA levels to achieve reduced BPA concentrations to the pre-industrial levels or its complete elimination.

BPA may have an effect that needs to be deciphered from the existing data to prevent its long-term negative impact. As once, unaware of the health risks, Roman populations had been poisoned gradually by an increased lead content in the water, through utilization of leaded pipes in their water distribution network (*Delile et al., 2014*), which subsequently contributed to the decline of Roman empire, the environmental and health toll of BPA plastics in the human environment needs to be addressed thoughtfully in the modern world. Further experimental and epidemiological efforts are necessary to fully establish a magnitude of potentially hazardous effects of BPA in humans, and its association to insulin resistance and diabetes, as well as other human diseases.

### Funding

The author received no funding for this work.

## Competing Interests

The author declares there are no competing interests.

## Author Contributions

- Milos Pjanic wrote the paper, prepared figures and/or tables, reviewed drafts of the paper.

## Data Availability

The research in this article did not generate any data or code (literature review).

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
