# Peer review of "The role of polycarbonate monomer bisphenol-A in insulin resistance"

_PeerJ, doi:10.7717/peerj.3809_

## Round 0.1 · original submission · Major Revisions

· Academic Editor

Major Revisions

Dear Dr. Pjanic,

Your manuscript entitled " Role of polycarbonate monomer bisphenol-A in insulin resistance" which you submitted to PeerJ, has been reviewed by the editor and 2 experts in the field.

I regret to inform you that the reviewers have raised some significant concerns that need to be addressed before the manuscript can be considered further. Since the reviewers do find some merit in the paper and the topic of the review is of broad and cross-disciplinary interest, I would be willing to reconsider if you wish to undertake major revisions and resubmit.

If you decide to resubmit the revised version, please summarize all the improvements made in the new version and give answers to all critical points raised in the reviewers’ report in an accompanying letter. Since the review article focus on the role of BPA in insulin resistance and diabetes, the proposed molecular mechanisms at play in this process should be discussed. Also the physiopathology of insulin resistance and glucose intolerance in general and, particularly, in pregnancy, deserves to be better and more thoroughly addressed. Finally, for completeness and usefulness of information provided, the section on sources of BPA exposure should be expanded.

Reviewer 1 ·

Basic reporting

This review focus on the topic of BPA and insulin resistance however it does not make a clear distinction between the effects on the mother and the offspring in terms of glucose homeostasis and insulin sensitivity. Overall bibliography is scarce and some important references of animal and epidemiological studies that connect BPA and metabolic disorders are lacking. Some general concepts like the role of estrogens in glucose homeostasis or the implications of gestational diabetes and insulin resistance during pregnancy are not exposed with sufficient clarity. Although the topic of the review is of interest major revision should be done. Please find below some of the major points I would suggest to correct in order to improve the manuscript.

Experimental design

Not applicable since it is a review

Validity of the findings

I would not valorate this since this is a review not a experimental study. Nevertheless the topic of the manuscript is of important interest.

Additional comments

1. Introduction: Other sources of BPA exposure of important relevance like thermal paper should be commented.

2. In some parts of the manuscript (lines 119, 231) it is mentioned that BPA is considered a weak estrogen. This should be clarify since BPA has been demonstrated to have a lower affinity for ERs than E2 when acting through the classical pathway as a transcriptional factor but different studies have demonstrated that BPA can promote estrogen-like activities with the same potency than E2. These mainly are the result of the activation of rapid responses via non-classical ER pathways or the different BPA recruitment of co-activators or co-repressors. This has been commented in Welshons et al, Endocrinology 2006; Alonso-Magdalena P et al Mol Cell Endocrinol 2012.

3. Line 185 The concept of gestational diabetes should be redefined since GDM is defined as any degree of glucose intolerance that it is first recognized during pregnancy. This is not clearly stated in the review. In addition the long-term consequences of GDM in the mother should be commented.

The concept and importance of insulin resistance during gestation should be better explained. Insulin resistance occurs in a physiological manner during gestation in order to guarantee the proper nutrients supply to the growing fetus. To compensate for that endocrine pancreas adapts by increasing pancreatic -cell mass and insulin release. When this compensation fails then hyperglycemia appears and the GDM is diagnosed. The problem with BPA exposure is that it can aggravate insulin resistance in the mother with the concomitant detrimental metabolic consequences for her.

In the present form of the manuscript these basic concepts are difficult to follow.

Surprisingly the effects of BPA on glucose metabolism in the mother are not commented. Some of the papers focused on that are Alonso-Magdalena et al, EHP 2010; Susiarjo et al Endocrinology 2015; Alonso-Magdalena et al, Endocrinology 2015.

4. In keeping with this, the role of estrogens on glucose homeostasis needs a better approachment. I recommend the authors to read some of the revisions written by Mauvais-Jarvis´s group or Nadal´s group. It is well known that estrogens have a positive role on glucose homeostasis within a narrow physiological window outside that it can promote insulin resistance. Although this concept is developed in the section “role of estrogen in gestational diabetes insulin resistance” I would suggest trying to improve it.

5. The effects of gestational BPA exposure on the offspring in terms of glucose metabolism are well known. The first demonstration that low doses of BPA exposure during pregnancy provoked in the male offspring glucose intolerance, insulin resistance and altered pancreatic -cell function came with the paper Alonso-Magdalena et al, EHP 2010. Later on many other papers come out confirming the phenotype observed. The effects were depending on dose and gender. In addition the effects of HFD in BPA-exposed animals have been explored. Some examples are Garcia-Arevalo et al Plos One2014; García-Arevalo et al, 2016 Endocrinology (already commented); Wei et al, Endocrinology 2011; Liu et al, Plos One 2013; Angle et al, Reprod Toxicol 2013; Li et al, Toxicol Letter 2014; Ma et al, Diabetologia 2013; Jiang et al, Toxicology letters 2014.

All these other studies are lacking in this review. Importantly the disrupting effects on glucose and lipid metabolism have been described in male but also in female offspring in some studies. This information should be added and correctly discussed.

6. The differences in the phenotype reported are in part related to the non-monotonic dose response of BPA. This concept has to be defined and explained in detail since it is a very important phenomenon in the EDC mode of action.

7. Some epidemiological studies in which BPA and diabetes are connected are also lacking.

8. Abstract should be rewritten in order to give a clearer and more direct message to the reader.

Reviewer 2 ·

Basic reporting

The manuscript is clearly written. It is a nice and challenging compendium on what is known and what can be done to improve the scientific knowledge on the role of BPA upon in utero exposure. Further, all concepts are easily explained and the background/context is usually properly provided (see below). As a review, the figures are sufficient and self-explaining and the discussion properly summarize the whole text content.

Literature references are also pertinent, although the Reviewer noticed the usefulness of some improvements to the background information and/or discussion regarding the following three aspects:

1. is there any further literature info the author could add on the role of BPA on hepatocytes that could be of interest to the manuscript topic ? The reviewer is refering mainly to the pre--post-natal hepatic switch from fetal to adult hepatoctytes;

2. since some data are available, could the Author expand the info on BPA release from plastics also to cans and orthodontic materials ? It will help the reader to consider the different sources of exposure to BPA. In this context, a few details also to the BPA presence and release from thermal paper will improve the manuscript;

3. the Author mentioned the estrogen-like role of BPA mediated by interaction with three nuclear receptors (ER-alpha and -beta, and ERR-gamma): does the Author considered the knowledge that ER-alpha and IGF-1 signalling pathways are cross-talking? Could the Author consider such interaction within the context of the gestational diabetes mellitus and the related insulin resistance ?

Experimental design

As a review, the literature search methods are well described. As mentioned above the literature search terms can be expanded to answer the three above suggested aspects.

Validity of the findings

The manuscript fulfils this section except for the aspects already mentioned above.

Additional comments

As already mentioned, the Reviewer found really interesting the manuscript and hope the suggestions will encourage the Author to improve it with some details that will contribute to its completeness.

As a minor points, the Reviewer ask also the following:
- delete the trait between Bisphenol-A at lines 1 (title), 21, 36, 48, 58 in order to get Bisphenol A;
- Line 26: write "Endocrine-Disrupting Chemical" instead of "endocrine disruptor chemical";
- Line 48: the firs time in which is mentione BPA I will include also its IUPAC name:
- Line 118: write "its" instead of "it's";
- Line 120: write "receptors" instead of "receptor";
- Line 121: abbreviate Dalton as "Da" in both cases:
-

---

## Round 0.2 · Minor Revisions

· Academic Editor

Minor Revisions

Dear Dr. Pjanic,

Thank you for your resubmission. Our referees have now considered your paper and have recommended publication in PeerJ. However, Reviewer 2 suggested some typographical changes and I think it would be appropriate to address these comments before acceptance. Please consider the suggestions of Reviewer 2. Otherwise, this manuscript is suitable for publication in PeerJ Journal.

Reviewer 1 ·

Basic reporting

The author has answered all the points raised by this reviewer

Experimental design

NA

Validity of the findings

NA

Additional comments

The author has answered all the points raised by this reviewer

Reviewer 2 ·

Basic reporting

No comment

Experimental design

No comment / Not applicable

Validity of the findings

No comment

Additional comments

The Author revision strongly improved the completness of the manuscript topic. I suggest a careful editing work and a careful check on the use of acronyms, symbol font and hyphens to be performed during the draft check. Furthermore, some references within the list must be corrected and others, those ones to institutional documents of EFSA and TIB, should be reported differently following the rules of the Editor.

As examples of the above mentioned checks to be done:

1. Acronyms:
- In the newly added paragraph "Role of estrogens in gestational insulin resistance", the acronym E2 is newly explained by its extended name: E2 has been already introduced and used several times before this point. Furthermore, in some cases is not necessary to use the extended name and it could be easily substituted by the acronym.
- At the end of the paragraph "Molecular mechanisms of BPA in promoting endocrine disruption, gestational insulin resistance and diabetes mellitus", the acronym LRRX is not understandable: the Reviewer doubts about it. It might be better to explain the concept without any acronym use.

2. Symbol font:
- transfrom all the gene names containing alpha (e.g. ER.alpha) and so on in ER-a using the symbol font for the a letter (probably will be done by the Editor during the draft process).

3. Hyphens:
- Withinthe paragraph "Molecular mechanisms of BPA in promoting endocrine disruption, gestational insulin resistance and diabetes mellitus" appear a hyphen between the words "molecular-level": to be deleted.

4: References:
- Aekplakorn W., et al. 2015 contains Asiatic letters to be deleted.
- Wynn V., et al. 1979 has the manuscript title all in capital letters and should be fixed correctly.
- Polycarbonate A Techno Commercial Profile Part 1 - Technische Informationsbibliothek (TIB) cannot be mentioned as it is now along the text and within the reference list.
- Bisphenol A | European Food Safety Authority. Available at http://www.efsa.europa.eu/en/topics/topic/bisphenol (accessed June 23, 2017) cannot be mentioned as it is now along the text and within the reference list.
- Bisphenol A (BPA): 2017 World Market Outlook and Forecast up to 2021 cannot be mentioned as it is now along the text and within the reference list.

---

## Round 0.3 · accepted · Accept

· Academic Editor

Accept

Dear Dr. Milos,

Thank you for submitting a revised version of your manuscript. I am pleased to inform you that your manuscript is accepted for publication in PeerJ in its current form and will now be forwarded to the product editor for copy editing and publication.

Yours sincerely

Stefano Menini